# Comparative Study of Anti-Corrosion Properties of Different Types of Press-Hardened Steels

**DOI:** 10.3390/ma17051022

**Published:** 2024-02-23

**Authors:** Hao Peng, Yunlong Zhao, Wanwan Fu, Zhishan Chen, Man Zhang, Jiesheng Liu, Xiaoming Tan

**Affiliations:** School of Civil Engineering and Architecture, Wuhan Polytechnic University, Wuhan 430023, China; zyl20000529@sina.com (Y.Z.); whwanwanfu@whpu.edu.cn (W.F.); 13018039076m@sina.cn (Z.C.);

**Keywords:** corrosion resistance, press-hardened steel, Al–Si coating, CSP

## Abstract

Hot stamping (or press hardening) is a new technology that is widely used in the production of advanced high-strength steel parts for automotive applications. Electrochemical measurements, including potentiodynamic polarization and electrochemical impedance spectroscopy (EIS), and accelerated corrosion tests (the neutral salt spray test and periodic immersion test) were conducted on press-hardened samples produced from uncoated (cold-rolled and cold strip production (CSP) hot-rolled) and Al–Si-coated press-hardened steels to elucidate their distinct anti-corrosion mechanisms. The cross-sectional micromorphology and element distribution of three types of press-hardened steels after a neutral salt spray test were observed using scanning electron microscopy (SEM) and energy-dispersive X-ray analysis (EDAX). The corrosion resistance of Al–Si-coated press-hardened steel was found to be significantly diminished following the hot stamping process due to the presence of microcracks and elevated iron content in the coating subsequent to austenitizing heat treatment. On the other hand, the corrosion resistance of uncoated press-hardened sheets produced from cold-rolled and CSP hot-rolled press-hardened steel was found to be proximal due to their nearly identical composition and microstructure (fully martensite) after the hot stamping process. Considering the high efficiency and energy-saving properties of hot-rolled press-hardened steel, it holds the potential to replace cold-rolled and even aluminum–silicon-coated press-hardened steel in automobile manufacturing.

## 1. Introduction

Hot stamping (also called hot-press forming or press hardening) is a new technology that has been widely used in the production of advanced high-strength steel parts for automotive applications in recent years. After heating the steel blanks of press-hardened steel in a furnace for a few minutes to an austenitizing temperature (≈880~930 °C), they are quickly transferred to a press in which they are immediately formed and hardened simultaneously using die quenching. The resultant press-hardened automotive components have a fully martensitic structure and exhibit a tensile strength of up to 1500 MPa. By virtue of these merits, the parts obtained via this technology are mainly used as the anti-collision components of automobiles, such as bumper beams, door reinforcements, A- and B-pillar reinforcements, and so on [1,2,3,4,5,6,7,8,9]. 

Traditionally, press-hardened steels are manufactured from cold-rolled steel, Al-Si-coated steel, and hot-rolled steel, among which cold-rolled steel has the largest share. Cold-rolled steel sheets are produced by rolling hot-rolled coils below the recrystallization temperature (<1000 °C). As heating is not involved in the cold rolling process, common defects produced during the hot rolling process, such as pitting and oxide scales, are effectively eliminated. Moreover, the dimensional precision of cold-rolled products is high, so its comprehensive mechanical properties and processability are better. However, cold-rolled steel has the obvious disadvantage that the cold rolling process requires a higher rolling force and intermediate annealing steps to eliminate work hardening in the rolling process. The low rolling efficiency and complexity of the production process increase its production cost, resulting in the additional waste of resources and large amounts of greenhouse gas emissions. As for Al–Si-coated steel, even though Al–Si-coated press-hardened steel can prevent the surface oxidation of the steel blanks during the hot stamping process and the final parts can be directly painted without shot blasting, it demonstrates problems such as the presence of microcracks and high iron concentration in the coating after hot stamping, no cathodic protection ability, a high cost, and so on. Additionally, its corrosion resistance shows a huge decrease after the hot stamping process [1,10,11,12,13,14,15]. Since few studies on Al–Si-coated steels in the field of press-hardened steels have been carried out, our team conducted relevant experimental studies on them in terms of anti-corrosion properties and proposed corresponding guidelines for their application in press-hardened steels. At present, there are few studies on the transformation of Al–Si coatings at high temperatures [16,17,18]. In order to explore the change in their corrosion resistance at high temperatures, we carried out corresponding tests. It was found that the corrosion resistance of Al–Si coatings significantly decreased after the hot stamping process, and the reason for this phenomenon was analyzed.

Comparatively, the manufacturing process of hot-rolled steel involves heating a billet to a temperature above the material’s recrystallization temperature (>1000 °C), followed by continuous rolling, extrusion, forging, and production. Throughout this entire process, both the shape and size of the material undergo significant changes [19,20,21,22,23,24]. Hot-rolled steel offers advantages such as relatively easy rolling and a high rolling efficiency. The controllable hot rolling temperature encompasses both the initial rolling temperature and finish rolling temperature. The initial rolling temperature is typically set at around 80% of the solid phase line temperature indicated in the alloy phase diagram, while the finish rolling temperature is generally regulated to be above the recrystallization temperature in the alloy [25,26].

Compact strip production (CSP) serves as an economically efficient steel production technology, revolutionizing the hot strip production process. In comparison to traditional methods, CSP offers a shorter and low-carbon process for hot strip production. Minimizing the need for secondary heating processes enables faster production rates and facilitates the creation of thinner product thicknesses. Initially focused on processing low- and medium-carbon steel grades, the first CSP facility embarked on this transformative journey. Currently, it covers almost the entire hot-rolled product range, including press-hardened steels [27,28]. By means of the development of the relevant technologies for CSP, the production of press-hardened steel with the required microstructure and mechanical properties has been achieved. In this context, the use of CSP hot-rolled steel instead of cold-rolled steel (including Al–Si-coated press-hardened steel) in the production of press-hardened steels will become a popular trend in the future. Due to the continuously increasing need for press-hardened steels, especially in the automotive industry, the CSP hot strip can expect to attain clear growth [29,30]. The price of cold-rolled steel is about 3.6 times that of CSP hot-rolled steel, and the price of aluminum–silicon-coated steel is about 4.7 times that of CSP hot-rolled steel. Therefore, on the economic level, CSP hot-rolled steel has greater advantages. In order to investigate the anti-corrosive properties of this novel, environmentally friendly CSP hot rolling process [27,31], a series of corrosion resistance tests were conducted on CSP hot-rolled steel, conventional cold-rolled press-hardened steel, and Al–Si-coated press-hardened steel. A novel approach proposing the utilization of hot-rolled steel instead of cold-rolled steel for producing press-hardened steel was introduced, offering fresh perspectives and insights into energy conservation and emission reduction.

In many fields of application, especially in the field of automobile manufacturing, the durability of steel is significantly influenced by its corrosion resistance, and achieving optimal corrosion resistance ensures the enhanced applicability of steel. In order to comprehensively evaluate the corrosion resistance of three common types of press-hardened steels, this work compares the electrochemical properties of two types of uncoated press-hardened 22MnB5 steels (cold-rolled and CSP hot-rolled) with one based on 22MnB5 coated with an Al–Si coating following flat-plate quenching treatment. The microtopography and chemical composition of the corrosion products formed in neutral salt spray tests on the three types of press-hardened steels were also analyzed. Two questions, whether the production process of press-hardened steels (cold-rolled and CSP hot-rolled) affects the corrosion resistance of press-hardened parts (uncoated) and whether press-hardened Al–Si coatings have cathodic protection, were effectively answered. In this study, the levels of corrosion resistance of press-hardened sheets using cold-rolled and CSP hot-rolled press-hardened steels were also compared by means of electrochemical measurement and laboratory accelerated corrosion tests [32,33]. Given the distinct variations in the inclusion composition and surface roughness between the two uncoated steels, it was imperative to assess the feasibility of substituting cold strips with CSP hot strips for press-hardened steel production in terms of safety. By conducting a comparative analysis on the corrosion resistance of three types of press-hardened steel samples following the hot stamping process, this study explored the feasibility of substituting cold-rolled steel with hot-rolled steel. It was observed that the corrosion resistance of Al–Si-coated press-hardened steel, despite its higher cost, significantly diminished after undergoing the hot stamping process. The underlying reasons for this phenomenon were thoroughly examined and discussed herein. These findings offer valuable insights for promoting sustainable development in press-hardened steel.

## 2. Experimental Section

### 2.1. Materials

Sodium chloride (NaCl, AR, Sinopharm Chemical Reagent Co., Ltd., Shanghai, China) and sodium bisulfite (NaHSO_3_, AR, Sinopharm Chemical Reagent Co., Ltd., Shanghai, China) were used as corrosive media in the accelerated corrosion tests (the neutral salt spray test, periodic immersion test, and electro-chemical measurements) of the test samples. Anhydrous ethanol (C_2_H_6_O, AR, Sinopharm Chemical Reagent Co., Ltd., Shanghai, China) was used to clean the specimens. All the press-hardening sheets used in the present study (uncoated cold-rolled and CSP hot-rolled steel, Al–Si-coated steel) were sampled in the form of blanks from industrially produced coils with the same thickness (1.50 mm). In the accelerated corrosion tests, an electronic analytical balance (BSA, Sartorius Scientific Instruments (Beijing) Co., Ltd., Beijing, China) with an accuracy of 0.1 mg was used to weigh the test samples.

### 2.2. Sample Preparation

Three kinds of steel blanks were press-hardened by plate quenching in order to simulate the actual hot stamping process of automobile parts. All the blanks were heated in a furnace to 930 °C and kept for 300 s, and then transferred to the plate quenching die for stamping and immediately quenched to room temperature after the die cooled. The two uncoated steel samples (cold-rolled and CSP hot-rolled steels) needed to be shot-blasted due to the covered scales on their surfaces after hot stamping, while no shot blasting was performed for the Al–Si-coated sample. For electrochemical measurements, the size of all the samples was 10 mm × 10 mm (L × W). For the laboratory accelerated corrosion tests (neutral salt spray test and periodic immersion test), the size of each sample was 100 mm × 50 mm (L × W). The test samples were degreased according to the specifications before all the tests were carried out. The detailed degreasing process consisted of washing the specimens with anhydrous ethanol followed by deionized water in sequential order and then drying.

The specifications of three kinds of press-hardened steel samples and the main processing parameters of plate quenching are shown in Table 1 [9].

### 2.3. Electrochemical Measurements

The electrochemical measurements were conducted using a commercially available 1280C electrochemical analyzer/workstation (Solartron Analytical, Shanghai, China). A three-electrode system was employed, consisting of a press-hardened steel sample with a surface area of 1 cm^2^ (cf. 2.2) as the working electrode, a platinum sheet as the counter electrode, and a saturated calomel electrode (SCE) as the reference electrode. All potentials were referenced to the SCE. The test solution was a 3.5 wt.% NaCl solution, which was prepared with an analytical-grade reagent and deionized water. The measurements were performed in a static solution at 25 °C, with polarization curve measurements carried out at an open circuit potential (OCP) ranging from −200 to +500 mV and scanned at a rate of 0.5 mV/s. The EIS was conducted under open circuit potential conditions (without polarization potential imposed), utilizing a frequency range of 10^−2^ to 10^4^ Hz and applying a sinusoidal disturbance with an amplitude of 5 mV.

For the polarization curve and EIS measurements, all the tests were performed in triplicate. The fitting program (SAI) used was provided by the electrochemical workstation (1280C electrochemical analyzer/workstation, Solartron Analytical Station).

### 2.4. Accelerated Corrosion Tests

Before the accelerated corrosion tests, the two uncoated steel samples (cold-rolled and CSP hot-rolled steels) needed to be shot-blasted (hung and passed through the industrial shot blasting line) due to the covered oxide layers on their surfaces after hot stamping, while no shot blasting needed to be performed for the Al–Si-coated sample. Then, the test samples were carefully cleaned with anhydrous ethanol followed by deionized water in sequential order and then drying.

The neutral salt spray test (NSST) was carried out using commercial model Q-FOG CCT 600 Cyclic Corrosion test equipment (Q-lab USA, Inc., Westlake, OH, USA, 640 L volume) under near 100% humidity conditions according to ASTM B117-03 [34] (Standard Practice for Operating Salt Spray Apparatus) testing requirements. The temperature of the exposure zone of the salt spray chamber was maintained at 35 ± 2 °C, and the samples were supported at the same angle (≈20°) from the vertical. The test salt spray was an atomized 5.0 ± 1.0% NaCl solution, which was prepared with an analytical-grade reagent and deionized water.

The periodic immersion test (wet/dry cyclic corrosion test) was carried out using commercial test equipment (Huaian Central Asia Equipment Co., Ltd., Huaian, China) consisting of the sample holders assembled on a rotating disk, solution tank, and heating lamp. The solution for the test was a (1.0 ± 0.5) × 10^−2^ mol/L NaHSO_3_ solution. The temperature of the test chamber was maintained at 45 ± 2 °C, and the humidity was maintained at 70 ± 5% RH. The period of each cycle was 60 min (including 12 min of immersion).

The periods of exposure were the same for the two accelerated corrosion tests: 24, 48, and 72 h.

Equations (1) and (2) were used to calculate the corrosion rate (CR) in mm year^−1^ [35] of the samples after the accelerated corrosion tests:∆*W* = *W*_1_ − *W*_2_(1)
where ∆*W* is the weight loss (g), *W*_1_ is the initial weight of the sample before accelerated corrosion tests (g), and *W*_2_ is the final weight of the sample after accelerated corrosion tests and pickling (g).
(2)$$CR(mm\;year^{-1})=\frac{∆W\cdot K}{\rho \cdot A\cdot t}$$
where ∆*W* is the weight loss (g) obtained in Equation (1), *ρ* is the metal density (g cm^−3^), *A* is the area of the sample (cm^2^), *t* is the exposure time (hours), and *K* is a constant depending on the unit of measurement of the corrosion speed according to ASTM G1-90 (1999 [35]) (=87,600).

### 2.5. Morphology and Element Distribution

The morphology and element distribution of the formed corrosion products for these three kinds of press-hardened samples during the NSST were analyzed using a QUANTA FEG450 field emission scanning electron microscope (SEM, FEI, Valley City, ND, USA) coupled with an EDAX energy dispersive spectrometer. In order to better observe the metallographic inlays, surface oxidation products, coatings, and steel substrates of the samples, a cross-section of each sample was analyzed. Before observation, in order to observe the morphology more clearly, the surfaces of the test samples after the NSST test were rinsed with deionized water and dried, after which the sample cross-sections were subjected to morphological and elemental analyses.

## 3. Results and Discussion

### 3.1. Electrochemical Measurement (Potentiodynamic Polarization Curves)

The potentiodynamic polarization curves of three types of press-hardened steel samples (cold-rolled, CSP hot-rolled, and Al–Si-coated) are shown in Figure 1. The electrochemical parameters, including the corrosion potential E_corr_ and corrosion current density (I_corr_), were obtained by performing a least-square fitting of the measured data in the weak polarization region and are listed in Table 2.

It can be seen from Figure 1 and Table 2 that the electrochemical behaviors of the two uncoated samples (cold-rolled and CSP hot-rolled) in the 3.5 wt.% NaCl solutions were similar. The values of E_corr_ and I_corr_ of the two uncoated samples were in the same order of magnitude. This showed that the difference in corrosion resistance between the two was not obvious. The similar electrochemical behaviors and close values of E_corr_ and I_corr_ observed in the two uncoated press-hardened steels could be attributed to their nearly identical chemical compositions and microstructures, characterized by a complete martensitic phase following austenitizing heat treatment and quenching during the hot stamping process. However, from Table 2, it can be observed that the value of the I_corr_ of the Al–Si-coated steel sample (1.813 × 10^−6^ A·cm^−2^) was much lower than that of the uncoated samples, indicating the better corrosion resistance of the Al–Si-coated sample. Meanwhile, it can be found from Figure 1 that the polarization curve of Al–Si-coated steel exhibited a distinct downward trend without an electrochemical passivation zone during the descent. This decrease could be attributed to the physical barrier provided by the Al–Si coating, which effectively impeded the diffusion of corrosion media into the steel base. The absence of an electrochemical passivation zone for Al–Si-coated steels illustrated that the Al–Si coating did not offer cathodic protection capability.

### 3.2. Electrochemical Impedance Spectroscopy (EIS) Measurements

To further compare the electrochemical behaviors of CSP hot-rolled and cold-rolled press-hardened steels and to evaluate the feasibility of replacing cold-rolled steel with CSP hot-rolled steel, we conducted an electrochemical impedance test (EIS) on both of the uncoated press-hardened steels. Nyquist plots for the press-hardened sheets produced from cold-rolled and CSP hot-rolled press-hardened steels are presented in Figure 2.

Both Nyquist plots display a depressed capacitive semicircle at whole frequencies, which could be attributed to the charge transfer process. To quantify the electrochemical impedance spectroscopy measurements of the electrochemical parameters, an equivalent circuit (R_s_(Q_dl_R_ct_)) was used for fitting the EIS data, where *R_s_* is the solution resistance and *Q_dl_* is the constant phase element (CPE). The CPE, which had a non-integer power dependence on the frequency, was used to represent the capacitance of the double layer accounting for the deviation from the ideal capacitive behavior due to surface inhomogeneity, roughness, and adsorption effects. *R_ct_* is the charge transfer resistance, indicating the corrosion resistance of the samples during the early stages. Table 3 lists the corresponding fitted electrochemical parameters.

It can be seen from the fitting results that the R_ct_ of the cold-rolled sample (992.80 Ω·cm^2^) was mildly higher than that of the hot-rolled sample (626.10 Ω·cm^2^), indicating the slightly better corrosion resistance of the cold-rolled sheet than that of the hot-rolled sheet (press-hardened) in the early stages of corrosion. Considering the similar corrosion resistance of CSP hot-rolled steel to cold-rolled steel, as well as its simple process and low cost, it is feasible to surmise that CSP hot-rolled steel could be utilized more often than cold-rolled steel in the future.

### 3.3. Accelerated Corrosion Tests

In this section, we conducted accelerated corrosion tests to compare the corrosion resistance of three types of unpainted press-hardened samples (cold-rolled, CSP hot-rolled, and Al–Si-coated). Figure 3 illustrates the surface appearances of these samples after undergoing the NSST for 9 h. As shown in Figure 3, following 9 h of NSST exposure, the surfaces of the cold-rolled and CSP hot-rolled samples were entirely covered with red rust, whereas the Al–Si-coated sample exhibited a relatively small area of red rust. This phenomenon showed that although the corrosion resistance of Al–Si-coated steel was better than that of cold-rolled steel and hot-rolled steel, all three of them showed rapid corrosion within a very short NSST span (9 h). The result was consistent with that of the electrochemical measurement (Section 3.1 and Section 3.2) and was due to the physical barrier protection of the Al–Si coating, which hindered the diffusion of corrosion media into the steel substrate [36]. However, it should be noted that the appearance of red rust on the surface of the Al–Si-coated sample after a short period of NSST exposure indicated that the press-hardened Al–Si-coated steel had a low level of corrosion resistance. This result was consistent with the electrochemical measurements (Section 3.1 and Section 3.2), showing that the Al–Si coating did not have any cathodic protection ability [9].

Figure 4 shows a comparison of the corrosion resistances of Al–Si-coated steel before and after the hot stamping process. From the figure, it can be seen that the Al–Si-coated press-hardened steel had excellent corrosion resistance before hot stamping as it still did not exhibit any corrosion destruction after 500 h of the NSST. After the hot stamping process, its corrosion resistance was remarkably reduced, which could be reflected in the large amount of rust produced on its surface during 9 h of NSST exposure. The results above indicated that the corrosion resistance of Al–Si-coated steel was better than those of cold-rolled and CSP hot-rolled steel after the hot stamping process, but it presented a significant reduction in corrosion resistance compared with that before the hot stamping process.

To further evaluate the corrosion resistance of two uncoated press-hardened sheets produced from cold-rolled and CSP hot-rolled steels, corrosion rate calculations during the NSST and periodic immersion testing were conducted on these two types of press-hardened steel samples. The corrosion rates of CSP hot-rolled and cold-rolled press-hardened steels in the NSST and periodic immersion tests were determined using Equations (1) and (2). Detailed data are compiled in Table 4 and Table 5. For the purpose of ensuring test accuracy, three sets of parallel samples were prepared. The corrosion rates of the press-hardened sheets produced from cold-rolled and CSP hot-rolled press-hardened steels during the NSST are shown in Table 4. It can be seen that the corrosion rates of both sheets decreased as the exposure time increased, which could be attributed to the formation of a stable and protective corrosion product layer during the NSST. However, the corrosion rate of the cold-rolled sheet was slightly lower than that of the hot-rolled sheet, indicating the better corrosion resistance of the cold-rolled sheet in a Cl^−^-containing environment. This was consistent with the potentiodynamic polarization curves and electrochemical impedance spectroscopy measurements (Section 3.1 and Section 3.2).

The corrosion rates of the press-hardened sheets made from cold-rolled and CSP hot-rolled press-hardened steels during periodic immersion testing are shown in Table 5. It can be seen that the corrosion rates of both steels were almost constant during the test, which could be attributed to the absence of a stable and protective corrosion product layer during wet/dry cyclic corrosion. However, the corrosion rate of the cold-rolled sheet was slightly lower than that of the hot-rolled sample, indicating the better corrosion resistance of the cold-rolled sheet. This was consistent with the results of the NSST, potentiodynamic polarization curves, and the electrochemical impedance spectroscopy measurements outlined above. The experimental results were the same as the electrochemical results, which once again underlined the feasibility of using hot-rolled steel rather than cold-rolled steel in the field of hot-formed steel.

Moreover, as indicated in Table 4 and Table 5, it was observed that the corrosion rate during the periodic immersion test was approximately four to five times higher than that of the NSST. This finding suggested a more rapid acceleration of corrosion for the periodic immersion test, possibly due to the usage of a more acidic solution (NaHSO_3_), an increased test chamber temperature, and the cyclic wet/dry conditions inherent in the periodic immersion test. Notably, the cyclic wet/dry conditions were found to have a detrimental impact on the corrosion product layer formed during the accelerated corrosion test. Furthermore, it is emphasized that the periodic immersion test could significantly reduce the required testing time for future studies.

### 3.4. Change of Properties of Al–Si-Coated Steel after Hot Stamping

To elucidate the underlying cause for the significant decline in corrosion resistance of Al–Si-coated press-hardened steel subsequent to hot stamping (cf. Figure 4), we also conducted an examination of its cross-sectional microstructures before and after the hot stamping process.

The cross-sectional morphologies of the Al–Si-coated steel before and after the hot stamping process are more clearly depicted at the microscopic level in Figure 5, showcasing notable alterations. Prior to hot stamping, a uniform and continuous Al–Si coating covering the steel substrate and a transition layer formed by diffusion in the production process are clearly visible. The continuous and compact coating and the cathodic protection of aluminum prevented the erosion of the corrosive medium on the steel substrate. However, following hot stamping, visible cracks and cavities emerged, thereby significantly reducing the corrosion resistance of the press-hardened Al–Si-coated steel. This structural transformation facilitated easier penetration of corrosive media through cracks and cavities. The formation of microcracks was attributed to the disparate coefficients of thermal expansion between the coating and steel substrate.

Additionally, we conducted an analysis of the chemical composition of the Al–Si coating following plate quenching (hot stamping) using EDAX. The measured quantities of aluminum (Al), silicon (Si), and iron (Fe) elements corresponding to the indicated regions in Figure 6 are presented in Table 6. Notably, in this study, it was observed that the iron concentration within the Al–Si coating after plate quenching reached as high as 44.2~62.1%. The enrichment of iron in the coating, resulting from the diffusion between the coating and steel substrate during austenitizing heat treatment, was responsible for this phenomenon. The brittle phase structures Fe_2_Al_5_, FeAl_3_, and FeSiAl_7_, etc., similar to ceramics, were formed on the surface [1]. The presence of microcracks and a high iron content in the coating contributed to the inadequate corrosion resistance observed in press-hardened Al–Si-coated samples, as depicted in Figure 3.

### 3.5. Microstructural and Component Analysis of Corrosion Products

Furthermore, by scrutinizing the cross-sections of the press-hardened sheets produced from CSP hot-rolled, cold-rolled, and Al-Si-coated steel using SEM, as shown in Figure 7, we compiled the corrosion depths of these three press-hardened samples following 9 h of NSST exposure (Figure 8 and Table 7). These data lent credence to the following assertions: in terms of the corrosion depth, both the maximum and average values, the corrosion depth of the Al–Si-coated sample was less than those of the cold-rolled steel and CSP hot-rolled steel, while the corrosion depths of the samples produced from CSP hot-rolled and cold-rolled steel were quite similar. This indicated that the corrosion resistance of Al–Si-coated steel after the hot stamping process was better than those of cold-rolled steel and hot-rolled steel, which had similar levels of corrosion resistance.

The cross-sectional microstructures of the three kinds of press-hardened samples (cold-rolled, CSP hot-rolled, and Al–Si-coated) after 9 h of NSST exposure are illustrated in Figure 7. It can be seen from Figure 7a,b that the formed corrosion products during the NSST of the two uncoated samples (cold-rolled and CSP hot-rolled) were mainly located on the steel substrate. The corrosion products generated on both uncoated samples during the NSST could be divided into an inner and outer layer. The energy dispersive analysis indicated that the corrosion products were mainly composed of Fe and O (not shown here). For the Al–Si-coated sample, shown in Figure 7c, the corrosion products were mainly located on the outside surface of the Al–Si coating and primarily consisted of Al, Si, Fe, and O elements according to the result of EDS (shown in Figure 6). The high iron content further substantiated the fact that the diffusion of Al–Si coating during the hot stamping process resulted in the exposure of a rigid matrix, consequently diminishing its corrosion resistance. Additionally, noticeable microcracks were observed within the Al–Si coating that longitudinally extended until reaching the inter-diffusion layer between the steel substrate and coating.

In summary, Al–Si-coated steel exhibited the best corrosion resistance among the three tested steels (Al–Si-coated steel, cold-rolled steel, and CSP hot-rolled steel) after the hot stamping process, but its corrosion resistance was greatly reduced after the hot stamping process, demonstrating a huge difference from that before the hot stamping process. Secondly, the corrosion resistance of cold-rolled steel was similar to that of hot-rolled steel, but it had no advantage compared with hot-rolled steel considering the price and preparation process, which is expensive to produce and has low production efficiency. In contrast, CSP hot-rolled steel is more economical and efficient because of its progress in the process without compromising the corrosion resistance. Accordingly, it is feasible to replace cold-rolled steel in the production and application of press-hardened steel with CSP hot-rolled steel now that more attention is being paid to green considerations such as emission reduction.

## 4. Conclusions

The following conclusions can be drawn:(1)In a chloride-containing environment, press-hardened sheets produced from Al–Si-coated steel demonstrated superior corrosion resistance compared to those produced from uncoated steels (cold-rolled and CSP hot-rolled) due to the physical barrier protection of the Al–Si coating, which hindered the diffusion of corrosion media into the steel substrate after plate quenching.(2)The corrosion resistance of Al–Si-coated press-hardened steel significantly diminished following the hot stamping process due to the presence of microcracks and elevated iron content in the coating subsequent to austenitizing heat treatment.(3)Both in Cl^−^- and HSO_3_^−^-containing environments, the corrosion resistance of the uncoated press-hardened sheet produced from cold-rolled and CSP hot-rolled press-hardened steel was found to be proximal due to their nearly identical compositions and microstructures (fully martensite) after the hot stamping process. Considering the high efficiency and energy-saving properties of hot-rolled press-hardened steel, it holds the potential to replace cold-rolled and even aluminum–silicon-coated press-hardened steel in some areas of application.

The automotive industry imposes stringent requirements on material corrosion resistance, and the durability of automobiles is significantly influenced by the corrosion resistance of automotive steel due to the unique service environment. With the increasing adoption of press-hardened steel in the automotive industry, it is imperative to conduct a more comprehensive investigation into the anti-corrosion performance and mechanisms of different types of press-hardened steel. Through our study on three types of press-hardened steel subjected to hot stamping, we clearly demonstrated the feasibility of substituting cold-rolled steel with hot-rolled steel. These findings provide valuable insights for advancing sustainable development in press-hardened steel.

## Figures and Tables

**Figure 1 materials-17-01022-f001:**
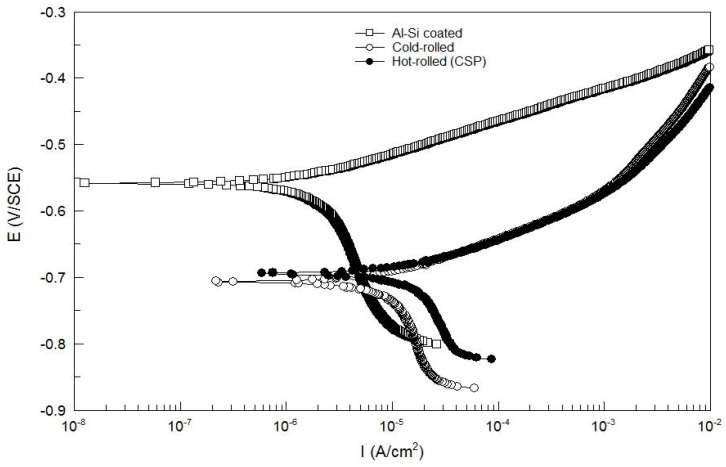
Potentiodynamic polarization curves of press-hardened steel samples (cold-rolled, CSP hot-rolled, and Al–Si-coated).

**Figure 2 materials-17-01022-f002:**
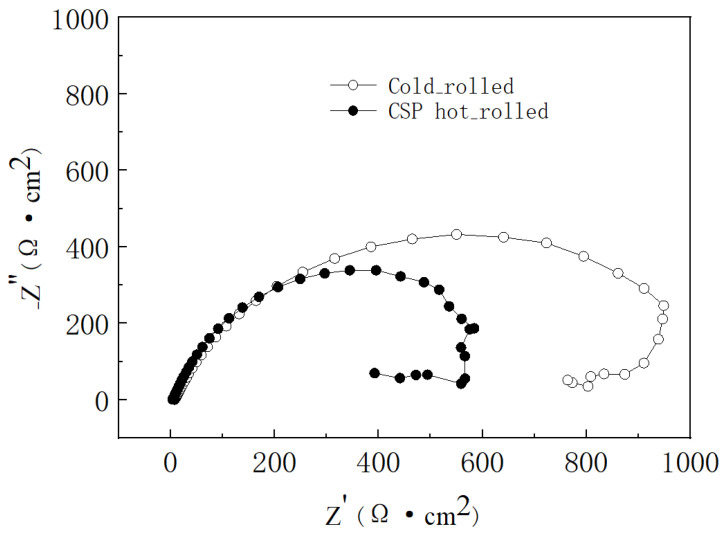
Nyquist plots of press-hardened sheets made from cold-rolled and CSP hot-rolled press-hardened steels in a 3.5 wt.% NaCl solution.

**Figure 3 materials-17-01022-f003:**
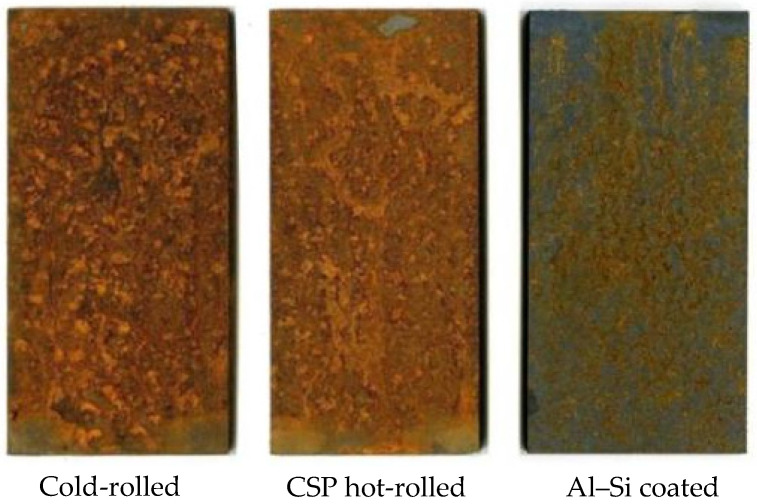
Surface appearances of unpainted press-hardened samples following the NSST for 9 h [9].

**Figure 4 materials-17-01022-f004:**
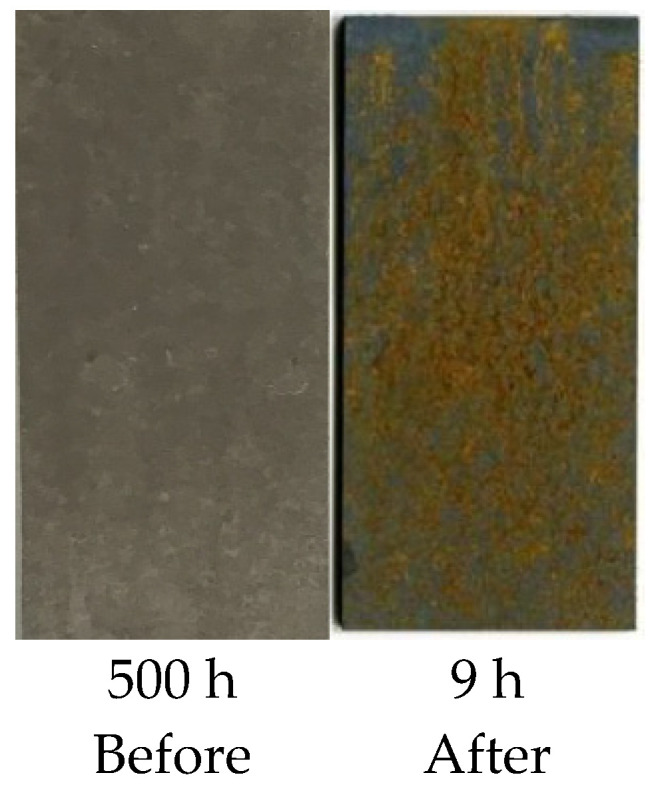
Surface appearances of the Al–Si-coated sample (after exposure to salt spray for a certain amount of time) before and after hot stamping process.

**Figure 5 materials-17-01022-f005:**
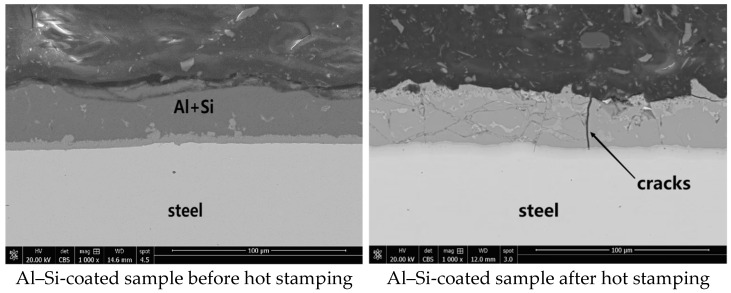
Comparison of Al–Si-layered steel before and after hot stamping process.

**Figure 6 materials-17-01022-f006:**
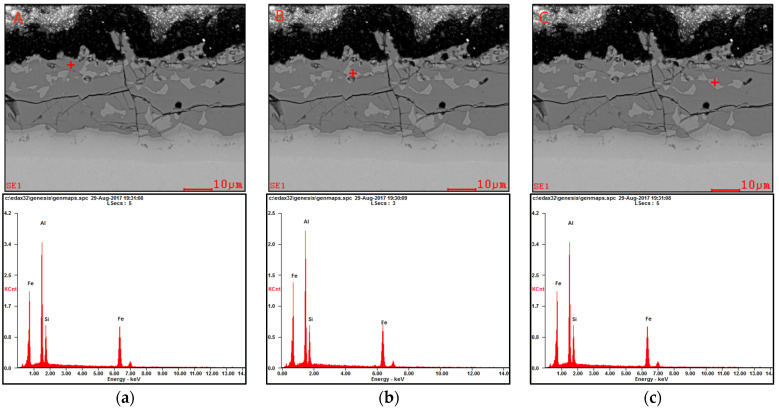
Cross-sectional microstructure and EDAX charts of the press-hardened Al–Si-coated sample. (**a**) Sample point A; (**b**) Sample point B; (**c**) Sample point C.

**Figure 7 materials-17-01022-f007:**
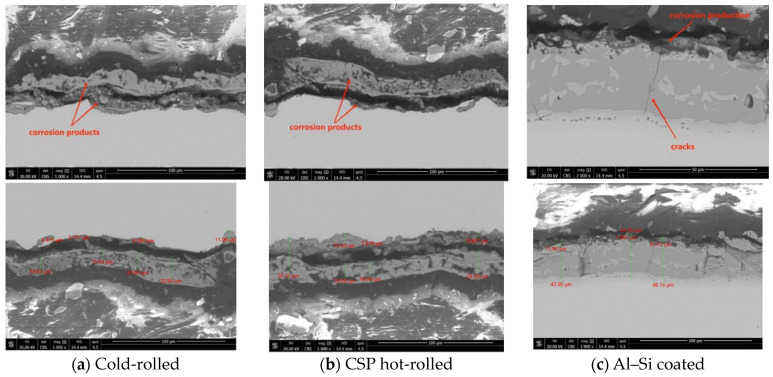
Cross-sectional microstructures of three kinds of press-hardened samples following 9 h of NSST exposure.

**Figure 8 materials-17-01022-f008:**
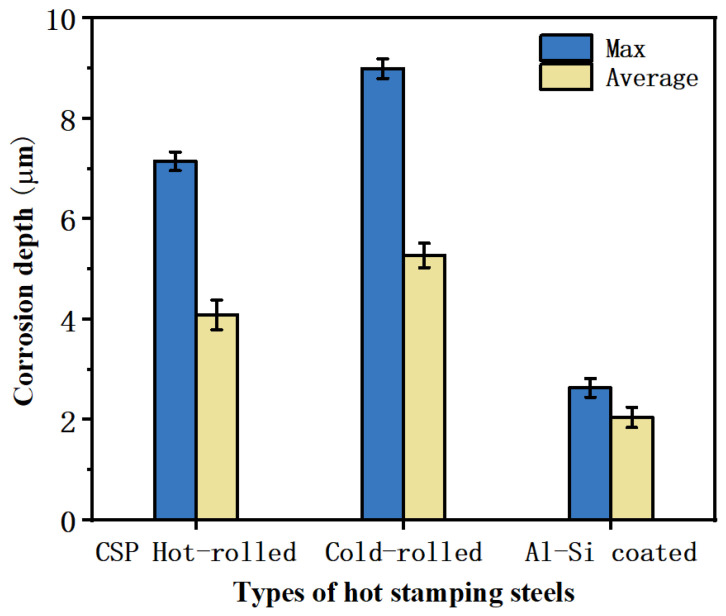
Corrosion depths of three kinds of press-hardened samples following the NSST for 9 h.

**Table 1 materials-17-01022-t001:** Test materials and main processing parameters of plate quenching [9].

Types of Press-Hardened Steels	Thickness (mm)	Heating Temperature (°C)	Heating Duration (s)	Press Pressure (t)	Holding Time (s)
Cold-rolled	1.50	930	300	230	15
CSP hot-rolled	1.50				
Al–Si coated	1.50				

**Table 2 materials-17-01022-t002:** Corrosion potential and corrosion current density of press-hardened steel samples in a 3.5 wt.% NaCl solution.

Sample	Corrosion Potential (vs. SCE)/V	Corrosion Current Density/(A·cm^−2^)
Cold-rolled	−0.706	1.332 × 10^−5^
CSP hot-rolled	−0.695	2.535 × 10^−5^
Al–Si coated	−0.558	1.813 × 10^−6^

**Table 3 materials-17-01022-t003:** EIS fitting results of press-hardened sheets made from cold-rolled and CSP hot-rolled press-hardened steels in a 3.5 wt.% NaCl solution.

Sample	*R*_s_/(Ω·cm^2^)	*R*_ct_/(Ω·cm^2^)
Cold-rolled	7.89	992.80
CSP hot-rolled	3.90	626.10

**Table 4 materials-17-01022-t004:** Corrosion rates of press-hardened sheets made from cold-rolled and CSP hot-rolled press-hardened steels during the NSST.

	Type of Sample					
	Cold-Rolled			CSP Hot-Rolled		
*W*_1_ (g)	56.364 ± 0.056	56.654 ± 0.078	56.294 ± 0.072	57.873 ± 0.066	57.751 ± 0.070	57.445 ± 0.071
*W*_2_ (g)	56.107 ± 0.066	56.156 ± 0.069	55.628 ± 0.059	57.528 ± 0.058	57.229 ± 0.077	56.792 ± 0.063
∆*W* (g)	0.257 ± 0.061	0.498 ± 0.072	0.666 ± 0.064	0.345 ± 0.062	0.522 ± 0.073	0.653 ± 0.067
*A* (cm^2^)	103.403 ± 0.087	102.637 ± 0.063	103.989 ± 0.072	104.270 ± 0.068	106.065 ± 0.077	104.288 ± 0.080
*t* (h)	24	48	72	24	48	72
*K*	87,600					
*ρ* (g cm^−3^)	7.85					
*CR* (mm year^−1^)	1.156 ± 0.047	1.12 8± 0.054	0.993 ± 0.042	1.538 ± 0.078	1.144 ± 0.065	0.970 ± 0.071

**Table 5 materials-17-01022-t005:** Corrosion rates of press-hardened sheets made from cold-rolled and CSP hot-rolled press-hardened steels during periodic immersion test.

	Type of Sample					
	Cold-Rolled			CSP Hot-Rolled		
*W*_1_ (g)	56.043 ± 0.072	55.983 ± 0.087	55.836 ± 0.077	57.476 ± 0.068	57.117 ± 0.059	57.530 ± 0.070
*W*_2_ (g)	54.940 ± 0.065	53.823 ± 0.082	52.273 ± 0.065	56.273 ± 0.078	54.575 ± 0.045	53.922 ± 0.066
∆*W* (g)	1.103 ± 0.069	2.160 ± 0.085	3.563 ± 0.069	1.203 ± 0.070	2.542 ± 0.052	3.608 ± 0.068
*A* (cm^2^)	104.563 ± 0.089	104.355 ± 0.092	104.109 ± 0.097	104.317 ± 0.064	104.493 ± 0.073	104.046 ± 0.090
*t* (h)	24	48	72	24	48	72
*K*	87,600					
*ρ* (g cm^−3^)	7.85					
*CR* (mm year^−1^)	4.905 ± 0.125	4.812 ± 0.089	5.304 ± 0.076	5.362 ± 0.144	5.656 ± 0.103	5.375 ± 0.123

**Table 6 materials-17-01022-t006:** Element content analysis of Al–Si coating after plate quenching using EDS.

Regions	Al (wt.%)	Si (wt.%)	Fe (wt.%)
A	52.68	3.13	44.19
B	28.31	10.95	60.74
C	27.96	9.92	62.12

**Table 7 materials-17-01022-t007:** Corrosion depths of three kinds of press-hardened samples following the NSST for 9 h.

Sample	Corrosion Depth (μm)			
	Max (Average)	Max (Variance)	Average (Average)	Average (Variance)
CSP hot-rolled	7.15	0.185	4.09	0.296
Cold-rolled	9.00	0.195	5.27	0.243
Al–Si coated	2.64	0.186	2.05	0.200

## Data Availability

Data are contained within the article.

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
