# Peer review of "Comparative Study of Anti-Corrosion Properties of Different Types of Press-Hardened Steels"

_materials, 2024, doi:10.3390/ma17051022_

Round 1
Reviewer 1 Report
Comments and Suggestions for Authors
The forthcoming publication is contingent upon the rectification of the comments. It is imperative to incorporate elemental mapping in elemental contrast onto the cross-sectional depiction of the specimens, thereby enhancing the elemental contrast.
A comprehensive structural analysis, such as X-ray Diffraction (XRD), should also be included to elucidate the underlying characteristics.
Figure 6 should incorporate an annotation delineating the margin of error inherent in determining corrosion thickness.
Furthermore, it is requisite to expound upon the incorporation of nitrogen into the Al-Si coating by the authors. From my standpoint, elucidation should be provided on whether the introduction of nitrogen mitigated the formation of cracks during deformation. It is essential to corroborate these findings with relevant data and analyses.
A citation to the pertinent work, namely "Nanomaterials-Based Coatings: Fundamentals and Applications" should be seamlessly integrated for scholarly transparency. This citation pertains to the volume encompassing pages 237 to 337, dated January 2019, and addresses the domain of hard and superhard nanostructured and nanocomposite coatings.
Comments on the Quality of English LanguagePlease, check the language of the manuscript once again after corrections.
Reviewer 2 Report
Comments and Suggestions for Authors
This work mainly focus on the Comparative Study of Anti-Corrosion Properties of Different 2 Types of Press-Hardened Steels. In this paper, the data is reasonable, the analysis is appropriate, it is suggested to accept after major modifications, the specific details are as follows.
Because there are some similarities with another article published by the first author in 2019 respectively:
H. Peng, X. P. Mao, X. Q. Huang, H. Wang, J. H. Song, T. Pang, Y. Ma, Z. Peng, and K. H. Hu, Comparative Study of Corrosion Resistance of Different Types of Press-hardened Steels,
Advanced High Strength Steel and Press Hardening, pp. 73-79 (2019), https://doi.org/10.1142/9789813277984_0012 presented at Conference: 4th International Conference on Advanced High Strength Steel and Press Hardening (ICHSU2018)
More details can be seen in the table below
|
Article published in 2019 |
Article send to Materials Journal |
|
|
|
|
|
|
|
|
|
|
|
|
1. I recommend that the authors to self-cite next to the table and figure that are already published (read more the Copyright and Permissions in the publication you already have the article published). There are publications that, if you are the author of an already published article and you want to use something from it, you don't need their acceptance, you just have to mention the source, there are others where you have to ask for permission.
1. The originality/novelty of the current work needs to be emphasized in the last few lines of the introduction.
2. In the introduction section you must add a sentence why it is important to study the phenomenon of corrosion.
3. Also should be cited in the introduction other articles that deal with this topic and highlight the novelty of the study.
4. There are some words written wrongly in the manuscripts, for example line 11 abstract ,, uncated” should be uncoated.
5. Line 28 introduction is ,, hardening steel in a furnace” written in bold.
6. Line 42 ,,processability are better.However....” after word better should be space.
the whole text must be analyzed because there are many phrases where a space must be inserted.
7. Experimental section at electrochemical measurements should be mentioned also the parameters for EIS , Frequency , AC , Freq/decade = 10, free (OCP) or potential imposed.
8. How many tests were done.... duplicate , triplicate. Should be mentioned with which program was fitting EIS and wich was the chi square value.
9. Must also be mentioned in section 2.4. how the samples were prepared before the immersion tests, more precisely how the native oxide layer formed in contact with air was removed.
10. Also with what kind of analytical balance was the weighing of the samples done.
11. All standard deviations must be put in the tables, and figures respectively Figure 6.
12. Table 4 and 5 should be completed with all the data how the results were reached )Ma, Mb, A and t.
Also with the indicated formula (equation 1), you have not calculated the corrosion rate, you calculated the weight loss.
For corrosion rate mm year−1 should apply another equation 
where ΔW is weight loss in grams obtained in equation 1 for all samples, 𝜌
is metal density in g cm−3, A is the area of the sample in cm2, t—is exposure time in hours, and K is a constant depending on the unit of measurement of the corrosion speed according to ASTM G1-90 (1999) more details in this article, also should be cited.
https://www.mdpi.com/1996-1944/14/16/4755
https://www.mdpi.com/2411-5134/8/1/39#B30-inventions-08-00039
13. The section conclusion must be extended
14. The references must be formated according to journal requirements

Round 2
Reviewer 2 Report
Comments and Suggestions for Authors
Thank you for accepting my suggestions. All required changes have been made.
The article can be accepted, after the authors format the references section in accordance with the journal's requirements.